# Classification Scheme of Heating Risk during MRI Scans on Patients with Orthopaedic Prostheses

**DOI:** 10.3390/diagnostics12081873

**Published:** 2022-08-02

**Authors:** Valeria Clementi, Umberto Zanovello, Alessandro Arduino, Cristina Ancarani, Fabio Baruffaldi, Barbara Bordini, Mario Chiampi, Luca Zilberti, Oriano Bottauscio

**Affiliations:** 1IRCCS Istituto Ortopedico Rizzoli, Laboratorio di Tecnologia Medica, Via di Barbiano 1/10, 40136 Bologna, Italy; valeria.clementi40@gmail.com (V.C.); cristina.ancarani@ior.it (C.A.); fabio.baruffaldi@ior.it (F.B.); barbara.bordini@ior.it (B.B.); 2Istituto Nazionale di Ricerca Metrologica, Strada delle Cacce 91, 10135 Torino, Italy; a.arduino@inrim.it (A.A.); m.chiampi@inrim.it (M.C.); l.zilberti@inrim.it (L.Z.); o.bottauscio@inrim.it (O.B.)

**Keywords:** MRI heating risk, MRI safety, orthopaedic implants, radiofrequency-induced heating, gradient-induced heating

## Abstract

Due to the large variety of possible clinical scenarios, a reliable heating-risk assessment is not straightforward when patients with arthroplasty undergo MRI scans. This paper proposes a simple procedure to estimate the thermal effects induced in patients with hip, knee, or shoulder arthroplasty during MRI exams. The most representative clinical scenarios were identified by a preliminary frequency analysis, based on clinical service databases, collecting MRI exams of 11,658 implant carrier patients. The thermal effects produced by radiofrequency and switching gradient fields were investigated through 588 numerical simulations performed on an ASTM-like phantom, considering four prostheses, two static field values, seven MR sequences, and seven regions of imaging. The risk assessment was inspired by standards for radiofrequency fields and by scientific studies for gradient fields. Three risk tiers were defined for the radiofrequency, in terms of whole-body and local SAR averages, and for GC fields, in terms of temperature elevation. Only 50 out of 588 scenarios require some caution to be managed. Results showed that the whole-body SAR is not a self-reliant safety parameter for patients with metallic implants. The proposed numerical procedure can be easily extended to any other scenario, including the use of detailed anatomical models.

## 1. Introduction

The growing number of implanted orthopaedic devices [1,2,3] has increased the need of MRI exams for patients carrying a metallic implant. The heating produced by MRI radiofrequency (RF) fields and, to a lesser extent, by gradient coil (GC) fields is a source of risk for passive non-magnetic metallic prostheses not fully regulated by the current safety standards [4,5,6,7]. Even if some clinical studies seem to downplay this problem by reporting the absence of patient discomfort or adverse events [8,9], the MRI scan of patients with arthroplasty should require some caution [10], according to the regulations in force. In this framework, several papers are addressed to verify the thermal safety of implantable devices in terms of specific absorption rate (SAR) and temperature increase [11,12,13,14,15,16].

Clinical MRI scanners rely on the whole-body averaged SAR or B_1_^+^ root mean square to limit the temperature increase in patients without metallic implants. Specific actions can be adopted for patients carrying “MR Conditional”-labelled implants to meet the implant specifications provided by manufacturers [10]. These labels [17] rely on standardized tests where the temperature rise is measured on the device positioned in a phantom and radiated by a defined electromagnetic RF field [6]. Such tests consider only a limited number of conditions, not always reflecting realistic clinical scenarios [11,12]. Furthermore, current tests are usually focused only on the RF field contribution, thus disregarding the thermal effects produced by the GCs, which are potentially non-negligible in bulky conductive prostheses [13]. Thus, currently, the clinical decision about the exam execution on patients with orthopaedic implants is taken on the basis of scientific literature, online available information [18,19,20], and personal professional experience.

This work proposes a general approach, made available on a public repository [21], to the assessment of the heating risk associated with MRI exams on patients with hip, knee, or shoulder arthroplasty. The analysis was performed through virtual experiments applied to a phantom similar to the one suggested by the American Society for Testing and Materials (ASTM), but the procedure holds for any setup, including detailed anatomical human models. The heating risk was evaluated in 588 clinical scenarios suggested by a frequency analysis of MRI exams on prosthetic patients. The results showed that less than 10% of the clinical scenarios should require some caution.

## 2. Materials and Methods

### 2.1. In Silico Evaluation of Heating Effects during the MRI Scans

In prosthetic patients, the heating induced by both the RF and GC magnetic fields is regulated by the Faraday’s law of induction. However, it occurs in complementary ways [14,22]. The RF energy is directly deposited inside the biological tissues. The metallic prosthesis modifies the RF field distribution, giving rise to possible hot-spots in proximity of the implant. The strongest thermal effects occur when the prosthesis is within the footprint of the RF antenna and when the implant’s longitudinal dimension is comparable to half wavelength [23].

The GC fields generate significant Joule losses only inside the metallic parts of the prosthesis. The induced heat then diffuses in the surrounding tissues. The highest heating is found when the implant experiences higher GC fields amplitudes, namely relatively far from the imaging region [24].

Specific numerical procedures were applied separately for the virtual RF and GC experiments [22]. In line with the International Electrotechnical Commission (IEC) standards [4], the assessment of RF-induced heating was conducted using SAR averages as a stress index. Differently, GC-induced heating was assessed looking at the temperature increase.

In the following, the term configuration identifies the coils, the subject and region of imaging, and the prosthetic implant. A scenario associates a given MR pulse sequence to a given configuration, identifying the specific case for which the heating risk assessment was desired.

### 2.2. RF Model

The procedure below applies to a generic single channel transmit RF coil, where the SAR deposited by the RF coil depends linearly on the incident power. Given a configuration *C* and a generic spatial coordinate x, the spatial distributions of the transmit sensitivity, ${\hat{B}}_{1,C}^{+}\left(x\right)$, and the SAR, ${\hat{\mathrm{SAR}}}_{C}\left(x\right)$, were computed in nominal conditions (i.e., a defined power incident to the RF coil).

A scaling factor $\beta_{C}$ was applied to ${\hat{B}}_{1,C}^{+}\left(x\right)$ to obtain a flip-angle $\alpha_{90{}^{\circ}}=90{}^{\circ}$ after an RF hard pulse of length Δt=1 ms. The flip-angle $\alpha_{90{}^{\circ}}$ was assumed to be reached when the free induction decay signal received by an ideal RF coil was maximised. According to this last condition, the proper value of $\beta_{C}$ was obtained solving the following maximisation problem:(1)$$\underset{\beta {}_{C}}{\max}\left|\int_{V}\sin\left(\beta_{C}\left|{\hat{B}}_{1,C}^{+}\left(x\right)\right|\gamma \Delta t\right)e^{i{\hat{\varphi }}_{C}^{+}\left(x\right)}dv\right|$$
where ${\hat{\varphi }}_{C}^{+}$ represented the phase of the transmit sensitivity, γ was the gyromagnetic ratio, and the integral was extended to a volume V denoting a central slab of the domain.

A given MR pulse sequence, *S*, was characterised by *N_S_* RF pulses in each repetition time (TR) of duration ${\mathrm{TR}}_{S}$. The *i*-th RF pulse in the TR was characterised by its duration $T_{S,i}$ and by the expected flip-angle $\alpha_{S,i}$. The SAR distribution in configuration *C* associated with the *i*-th RF pulse was
$${\mathrm{SAR}}_{C,S,i}\left(x\right)=r_{S,i}{\left(\frac{\alpha_{S,i}}{\alpha_{90{}^{\circ}}}\frac{\Delta t}{T_{S,i}}\right)}^{2}\beta_{C}^{2}{\hat{\mathrm{SAR}}}_{C}\left(x\right)$$

The coefficient $r_{S,i}$ accounted for the actual profile of the RF pulse modulation and was computed as the ratio between the energies associated with the modulated and hard RF pulses.

The SAR averaged over the TR was defined as
$${\mathrm{SAR}}_{C,S}\left(x\right)=\frac{1}{{\mathrm{TR}}_{S}}\overset{N_{S}}{\underset{i=1}{\sum }}{\mathrm{SAR}}_{C,S,i}\left(x\right)T_{S,i}=\underset{\psi_{S}}{\underbrace{\frac{1}{{\mathrm{TR}}_{S}}\overset{N_{S}}{\underset{i=1}{\sum }}\frac{r_{S,i}}{T_{S,i}}{\left(\frac{\alpha_{S,i}}{\alpha_{90{}^{\circ}}}\Delta t\right)}^{2}}}\underset{\xi_{C}\left(x\right)}{\underbrace{\beta_{C}^{2}{\hat{\mathrm{SAR}}}_{C}\left(x\right)}}=\psi_{S}\xi_{C}\left(x\right)$$

Thus, the SAR distribution produced in configuration *C* by sequence *S* was given as the product of a “sequence RF stress index” $\psi_{S}$, whose value depended only on *S*, and a “configuration index” $\xi_{C}\left(x\right)$, depending only on *C* and fully encoding the effect of different anatomies on SAR.

The spatial averages of SAR (over 10 g or the whole body) were deduced by simply averaging $\xi_{C}\left(x\right)$. The IEC 60601-2-33 provided SAR limits over 6 min ($T_{\mathrm{IEC}}$) of exposure. When the sequence duration $T_{S}$ was lower than $T_{\mathrm{IEC}}$, the $T_{\mathrm{IEC}}$ temporal average was computed as
(2)$${\mathrm{SAR}}_{C,S}^{T_{\mathrm{IEC}}}\left(x\right)=\frac{T_{S}}{T_{\mathrm{IEC}}}\psi_{S}\xi_{C}\left(x\right)$$

Limits were provided also for the average over 10 s. Most of the time, such a temporal window is shorter than the total sequence duration and longer than its TR. In this case, the 10 s temporal average was well approximated by the average on a TR, ${\mathrm{SAR}}_{C,S}\left(x\right)$. The distributions ${\hat{B}}_{1,C}^{+}\left(x\right)$ and ${\hat{\mathrm{SAR}}}_{C}\left(x\right)$ were computed with Sim4Life (Zurich Med Tech AG, Zurich, Switzerland) [25], assuming the perfect electric conductor approximation for the metallic components. A 50 mm thickness of the central slab, over which (1) was integrated, was chosen.

### 2.3. GC Model

The three GC coils represent a three-channel device. The time behaviour of the total gradient magnetic field was determined by the superposition of the gradient fields generated by each coil. For this reason, the deposited power cannot be related with the pulse sequence features in a way as simple as that proposed for the RF. However, the same Q matrix formalism [26], largely deployed with RF Multichannel Transmit Systems (pTx), can be extended to GC, leading to a rather simple relation between the power deposited inside an implant and each gradient coil signal.

Neglecting the skin effect within the metallic object [27] (i.e., adopting a conservative approximation), the current density vector J(x,f) induced in a spatial point x inside the analysed implant at the frequency  f was expressed as
$$J\left(x,f\right)=f\overset{3}{\underset{k=1}{\sum }}J_{k}\left(x\right)G_{k}\left(f\right)$$
where *k* identified the supplied gradient coil. $G_{k}\left(f\right)$ (expressed in Tesla per meter) was the complex harmonic contribution of the *k*-th coil at the frequency f and depended on the pulse sequence waveform. $J_{k}\left(x\right)$ was the current density vector induced in x at unit frequency (1 Hz) when the *k*-th gradient coil was supplied to generate a nominal gradient field with unit amplitude (1 T/m).

The square amplitude of J(x,f) was expressed as
$${\left|J\left(x,f\right)\right|}^{2}=J{\left(x,f\right)}^{H}J\left(x,f\right)=f^{2}\overset{3}{\underset{w=1}{\sum }}G_{w}{\left(f\right)}^{H}J_{w}{\left(x\right)}^{H}\overset{3}{\underset{k=1}{\sum }}J_{k}\left(x\right)G_{k}\left(f\right)$$
where H denoted the transpose conjugate. In matrix notation:$${\left|J\left(x,f\right)\right|}^{2}=f^{2}G{\left(f\right)}^{H}QG\left(f\right)$$
where G(f) was a three-element vector whose *k*-th element was equal to $G_{k}\left(f\right)$, and Q was a 3 × 3 Hermitian matrix whose *m,n* entry was given by $J_{m}{\left(x\right)}^{H}J_{n}\left(x\right)$.

The power density, p(x), in point x was
$$p\left(x\right)=\underset{f}{\sum }{\left|J\left(x,f\right)\right|}^{2}=\frac{1}{2\sigma \left(x\right)}\underset{f}{\sum }f^{2}G{\left(f\right)}^{H}QG\left(f\right)$$
where σ(x) was the electric conductivity in x. Finally, the power density averaged inside the implant volume $V_{P}$ was evaluated as
$$p=\frac{1}{V_{P}}\int_{V_{P}}p\left(x\right)dv=\frac{1}{2\sigma }\underset{f}{\sum }f^{2}G{\left(f\right)}^{H}\frac{1}{V_{P}}\int_{V_{P}}QdvG\left(f\right)=\frac{1}{2\sigma }\underset{f}{\sum }f^{2}G{\left(f\right)}^{H}Q_{V}G\left(f\right)$$

This expression represented a quadratic form analogous to that generally involved in pTx SAR computations at RF [26]. Here, the averaged power density p was given by a sum of terms including a contribution related to the pulse sequence, G(f), and another dependent on the configuration (i.e., gradient coil, region of imaging and implant), $Q_{V}$.

The following computational procedure allowed us to minimize the number of simulations required. For a given implant, three electromagnetic simulations, performed with a validated homemade finite element solver [24], provided the current density distribution inside the implant, $\hat{J_{i}}\left(x\right)$, induced at 1 Hz by a 1-T spatially uniform magnetic flux density directed along the *i*-th Cartesian axis. Denoted by $B_{k,i}$ the *i*-th component of the magnetic flux density generated by the *k*-th GC in the implant barycentre, the actual distribution $J_{k}\left(x\right)$ was:$$J_{k}\left(x\right)=\overset{3}{\underset{i=1}{\sum }}B_{k,i}\hat{J_{i}}\left(x\right)$$

The thermal problem was developed assuming a power density uniformly deposed inside the implant equal to 1 W/m^3^. This allowed performing a single thermal simulation for each implant, thus potentially extending the results to any scenario through a scaling factor. The computations were carried out in a domain including the implant and a suitable portion of media around it. The time evolution of the induced temperature increase $\hat{\vartheta }\left(x,t\right)$ was evaluated by solving the Pennes bioheat equation with a validated homemade code [28]. The time behaviour of the temperature increase produced in the domain by any sequence was then deduced from this single solution $\hat{\vartheta }\left(x,t\right)$, simply rescaling it by the average power density p coming from the electromagnetic problem: $\vartheta \left(x,t\right)=p\hat{\vartheta }\left(x,t\right)$.

A duty-cycle coefficient applied to the total power accounted for possible idle times within the sequence. For MR pulse sequences whose gradients vary during successive TR, the highest gradient intensity was kept in all TRs. This assumption simplified the harmonic content of the sequence leading to conservative results. The maximum temperature increase at the end of the sequence, ϑ, and at steady state, $\vartheta_{\infty }$, were used as GC stress metrics.

### 2.4. Frequency Analysis

A frequency analysis of MRI exams on patients carrying a hip, a knee, or a shoulder prosthesis was performed. Data were extracted from the Register of the Orthopaedic Prosthetic Implants (Registro degli Impianti Protesici Ortopedici, RIPO) [29], which collected data from 70 Orthopaedic Units integrated in the Italian National Health Service (Servizio Sanitario Nazionale, SSN) of Emilia-Romagna, an administrative region of Northeast Italy, with about 4.4 million inhabitants.

Collected data were cross-referenced with those recorded by two additional Italian regional healthcare system databases, Outpatient Specialist Assistance (Assistenza Specialistica Ambulatoriale, ASA) [30] and Hospital Discharge Form (Scheda di Dimissione Ospedaliera, SDO) [31]. All MRI exams in ASA and SDO were associated with an anonymized patient identifier. Features of RIPO, ASA, and SDO from June 2018 are presented in Appendix A.

RIPO data for 2013 (i.e., patients whose surgery was performed during 2013) were matched with the ASA and SDO, including only the patients subjected to at least one MRI scan in the following three years. The MR exam prescription code provided information about the region of MRI, without details about exam protocols or diagnostic questions. The analysis was limited to patients carrying only one kind of implant.

### 2.5. Scenario Selection

A homogeneous parallelepiped phantom (440 mm × 178 mm × 1190 mm), filled with the gel (electrical conductivity: 0.47 S/m, relative electric permittivity: 80) and prescribed by ASTM for measurement of induced heating [6], was used as the subject of imaging. The height of the phantom was enlarged with respect to the ASTM suggestions, so that the phantom could be kept centred within the scanner while moving the implant to reproduce different imaging-region conditions.

Four implant models, detailed in Figure 1, were analysed: one knee, one shoulder, and two hip models (denoted as A and B) with different sizes. The shoulder, knee and hip B implant models were provided by the relevant manufacturer. The model of the hip A implant was obtained through 3D scanning. In order to produce conservative results, all the metallic components were simulated as a CoCrMo alloy. Indeed, such an alloy produces higher temperature increases than those generated in titanium-based alloys (e.g., about 140% when a total hip implant is involved [24]) when exposed to the GC excitation. The implants were positioned within the phantom complying with the realistic location and orientation inside a human body. The longitudinal position was chosen consistently with the seven selected imaging regions shown in Figure 2.

The RF coil was a 16-leg shielded birdcage body coil (diameter: 713 mm, height: 450 mm), operating as a low-pass coil tuned to 64 MHz for the analysis at 1.5 T and as a high-pass coil tuned to 128 MHz for the analysis at 3 T. In both cases, the coil was supplied in quadrature operation mode. An apodised sinc modulation was assumed for all the RF pulse excitations.

For both 1.5 T and 3 T systems, traditional GCs for cylindrical bore scanners, arranged on a cylinder of radius 33.5 cm and height 150 cm, were used. The selected MRI pulse sequences (T2 FRFSE, T1 FSE, T2* GRE, 3D FSPGR, DWI SE-EPI, PERF GRE-EPI, and TrueFISP), which belong to four of the most common sequence families (spin echo (SE), gradient Echo (GRE), echo planar imaging (EPI), and balanced steady-state gradient echo (TrueFISP)), are presented in Table 1. Since all the examined sequences had a duration shorter than 6 min, the relation (2) was applied. Furthermore, the TR-averaged SAR can be considered representative of the 10 s averaged SAR, because all sequences had a duration considerably longer than 10 s and a TR far below 10 s.

Since, in the EPI sequences, the prevailing GC heating effects are due to the frequency encoding signal [28], the x-, y-, z-directions were explored. In total, 84 configurations (56 for RF and 28 for GC) were identified and associated with the seven MR pulse sequences previously indicated, leading to 588 investigated scenarios.

### 2.6. Heating Risk Classification

Three risk tiers were defined separately for the RF and GC fields, starting from the safety criteria in IEC 60601-2-33 [4]. For RF heating, the SAR limits prescribed by IEC were adopted. To account for the RF field perturbations produced by the implant, the limits relevant to local transmit coils were assumed, although a whole-body volume coil was deployed in the simulations. The IEC limits involve the whole-body SAR (${\mathrm{SAR}}_{\mathrm{wb}}$) and the maximum 10 g SAR (${\mathrm{SAR}}_{10g}$) with the following conditions
$$\{\begin{array}{c} {\mathrm{SAR}}_{\mathrm{wb},C,S}^{T_{\mathrm{IEC}}}\leq K_{1}\\ {\mathrm{SAR}}_{\mathrm{wb},C,S}\leq 2K_{1}\\ {\mathrm{SAR}}_{10g,C,S}^{T_{\mathrm{IEC}}}\leq K_{2}\\ {\mathrm{SAR}}_{10g,C,S}\leq 2K_{2} \end{array},$$
where $K_{1}=2\;W/\mathrm{kg}$ and $K_{2}=10\;W/\mathrm{kg}$ for the hip and the shoulder implants applied. For the knee implant, the threshold $K_{2}$ was doubled (20 W/kg).

An attention situation (risk tier 3) was associated with the scenarios exceeding at least one limit. A warning situation (risk tier 2) was associated with the cases where no previous limit was exceeded, but
$${\mathrm{SAR}}_{\mathrm{wb},C,S}>K_{1}{\;\mathrm{or}\;\mathrm{SAR}}_{10g,C,S}>K_{2},$$
namely, when a repetition or extension of the sequence might give rise to a risk tier 3. Finally, a safe situation (risk tier 1) was associated with all the other scenarios.

For GC heating, the limits were expressed in terms of temperature increase and were assumed to be 2 °C for the shoulder and hip implants, and 3 °C for the knee implants. These values originated from the 39 °C maximum local tissue temperature IEC limit^4^, assuming a 37 °C normal temperature in the trunk (including hip and shoulder) and 36 °C in the limbs (including the knee). Risk tier 3 was associated with the scenarios in which ϑ was above the threshold. Risk tier 2 was associated with the scenarios in which ϑ was below the threshold, but $\vartheta_{\infty }$ overcame it. Finally, risk tier 1 was associated with all other scenarios.

## 3. Results

### 3.1. Frequency Analysis

From the accounted database, 11,658 patients (34.3% males with an average age of 69.9 years, standard deviation of 11.5 years, range 17–101 years, and 65.7% females with an average age of 73.6 years, standard deviation of 10.4 years, range 13–104 years) underwent the surgery in 2013; 20.5% of them (34% males and 66% females) underwent at least one MR scan within three years from the time of surgery. Table 2 shows details on the type of prosthesis and categories of exams.

Head, spine, and musculoskeletal exams comprised over 90% of the MRI exams for all the patients carrying the three types of prostheses. The musculoskeletal exams included prosthesis-related studies, but also investigations on the other musculoskeletal joints. The spine exam categories ranged from cervical to lumbar regions.

### 3.2. RF Heating

A non-uniform Cartesian mesh, with a size equal to 1 mm in the implant and relaxed elsewhere in the phantom, was adopted. A discrepancy in the maximum value of the ${\mathrm{SAR}}_{10g}$ of less than 5% was found when comparing the result with a 0.5 mm mesh. For hip A, only a model discretized with 2 mm voxels was available.

The highest value of the whole-body configuration index $\xi_{\mathrm{wb},C}$, 6.13 W/kg, was reached for the hip B implant when imaging the femur at 3 T. The maximum RF stress index $\psi_{S}$ of the selected MRI pulse sequences was 0.12, reached by the TrueFISP sequences. Since the product between these maximum values was more than three times lower than the limit $K_{1}$, none of the investigated scenarios exceeded the RF whole-body SAR limits. Therefore, only the results on the local SAR averages are discussed in the following.

The two limits involving the local SAR average were combined in the single expression
$$\xi_{10g,C}\leq \frac{\min\left(2,\frac{T_{\mathrm{IEC}}}{T_{S}}\right)K_{2}}{\psi_{s}},$$
which determined a maximum value $\xi_{10g,S}^{*}$ of the local 10 g configuration index admissible for a sequence *S*, characterised by a duration $T_{S}$, and a RF stress index $\psi_{S}$. A map of $\xi_{10g,S}^{*}$ as a function of $T_{S}$ and $\psi_{S}$ is provided in Figure 3. When $T_{S}<T_{\mathrm{IEC}}/2$, the value of $\xi_{10g,S}^{*}$ depended only on $\psi_{S}$. The limits associated with the selected MRI pulse sequences are denoted by red dots in Figure 3.

Figure 4 shows the value of $\xi_{10g,C}$ for all the configurations where the implant falls within the RF coil. When the implant was positioned outside the RF coil, the values $\xi_{10g,C}=26.9\;W/\mathrm{kg}$ at 1.5 T and $\xi_{10g,C}=51.8\;W/\mathrm{kg}$ at 3 T were independent of the type of implant and similar to the case of the phantom without implant. The threshold values $\xi_{10g,S}^{*}$ for the risk tier 3 are reported in Figure 4 as solid horizontal lines. Dashed horizontal lines denote the threshold for the risk tier 2. In order to account for the higher limits with the knee implant, the configuration index was halved for the knee implant in Figure 4. This allows comparing all the $\xi_{10g,C}$ values with the reported thresholds. The risk tier 3 and tier 2 scenarios were collected in Figure 5.

Finally, the result of a sensitivity analysis of the SAR averages with respect to the RF pulse length is reported in Figure 6. The RF pulse length affected almost linearly the power deposition, with a variation of 30% in the SAR averages for a variation of 30% in the RF pulse length. This analysis was independent of the selected sequence and was performed keeping the time–bandwidth product of the pulse modulation constant.

### 3.3. GC Heating

The electromagnetic simulations used a uniform Cartesian mesh of 0.5 mm for all implants except hip A (2 mm as in RF simulations). The thermal properties of the phantom (thermal conductivity: 0.3 W/m/K, blood perfusion: 600 W/m^3^/K, density: 1000 kg/m^3^, specific heat capacity: 2500 J/K/kg) were set similar to those of bone tissue.

In most of the considered scenarios, the heating due to the GC magnetic fields was almost negligible. The risk tier 3 was not reached by any selected scenario. The occurrences of risk tier 2 are presented in Figure 5. According to a worst-case approach, for EPI pulse sequences, the highest GC-induced temperature increase, computed by setting the frequency-encoding direction along each Cartesian axis, was assumed. This value is reported in Figure 5, together with the frequency-encoding direction.

## 4. Discussion

### 4.1. Frequency Analysis and Selected Scenarios

The frequency analysis showed that most exams are carried out in the head, spine, and musculoskeletal targets, with a similar frequency for the studied prostheses. In clinical practice, the exams classified as ‘musculoskeletal’ are carried out for problems not solely related to prostheses. Even ascribing all the musculoskeletal exams to prosthesis-related studies, most MR exams were performed in regions not in proximity to the implant. This suggests that MRI exams are often required for clinical questions not related to the arthroplasty.

The results showed a wide variability in possible scenarios in daily clinical practice, justifying the application of the risk analysis to all the combinations between prostheses and imaging regions. In addition, in absence of standards for the definition of clinical MRI protocols and without specific information about the clinical motivations of the exams, this study was extended to a number of different MRI pulse sequences, whose parameters were defined as representative of those commonly used for the most frequent clinical questions.

### 4.2. RF Heating

The RF stress index $\psi_{S}$ showed a large range of variability (from 1.3 × 10^−5^ to 1.2 × 10^−1^), with the values for TrueFISP, T1 FSE, and T2 FRFSE sequences being higher with respect to the other ones. Therefore, these sequences have the lowest threshold $\xi_{10g,S}^{*}$, being the only ones with a safety threshold comparable to the configuration index of the selected configurations.

The configuration indexes at 3 T are generally higher than those at 1.5 T. The only exception was the hip B implant for pelvis or femur-imaging regions, presumably because of a particular combination of the implant size and the field wavelength [23]. In general, the highest configuration indexes were found for the imaging region centred within the RF coil. The shoulder prosthesis showed the highest configuration indexes, whereas the smallest ones were obtained with the knee implant. By comparing this maximum value of $\xi_{10g,C}$ (349 W/kg) with the RF stress indexes $\psi_{S}$ of the selected sequences, the scenarios involving the sequences T2* GRE, 3D FSPGR, DWI SE-EPI, and PERF GRE-EPI reached values of ${\mathrm{SAR}}_{10g,C,S}$ at least five times lower than the most severe limit for tier 2. Therefore, all the scenarios involving these sequences exhibited risk tier 1 for the RF-induced heating.

Only 29 out of 392 RF scenarios did not belong to risk tier 1. In all these scenarios, the whole-body SAR was within the safety range, thus potentially confirming that this quantity cannot be considered a self-reliant parameter to assess the safety of patients with metallic implants.

The sensitivity of the SAR averages with respect to the RF pulse length highlights that the discussed results hold exclusively for the selected sequence parameters. Furthermore, the linear dependence suggested how the RF-induced heating could be minimised by tuning the sequence parameters.

### 4.3. GC Heating

None of the 196 selected GC scenarios reached the risk tier 3. Only 21 scenarios with a risk tier 2 were found when executing the PERF GRE-EPI or the TrueFISP sequences. With these sequences, the tier 2 limit was surpassed in all the regions of imaging except for the head, for the hip A implant and, in a more limited number of occurrences, for hip B. As for RF, the knee implant always exhibited a risk tier 3. In some imaging areas, hip B produced higher temperature increases than hip A when performing the PERF GRE-EPI sequence, but for the TrueFISP sequence, the highest values were found in hip A. Such a result may prove that the separation between configuration and sequence, found for RF, may no longer apply for GC. Finally, the PERF GRE-EPI sequence, performed in imaging areas outside of the head, may not correspond to realistic clinical situations; nevertheless, it could represent a remarkable result from a conceptual point of view because the corresponding SAR values for RF were below the prescribed limits.

### 4.4. Limitations

The frequency analysis focused only on the population of Emilia-Romagna. Such a limitation is partially compensated by the fact that the RIPO data are fairly in line with the European ones [32] and by the completeness of the data about the involved orthopaedic implants (>95%).

In GC scenarios, magnetic flux density and deposed power were assumed to be uniformly distributed inside the implant. Direct comparisons between experiments and computations, performed under the same hypothesis, showed a discrepancy less than 10% in the temperature increase both after relatively short exposures (20 s) [33] and after longer exposures (15 min) [34]. In addition, Wooldridge et al. [23] showed that the total power deposed within a bulky orthopaedic implant correlated well with the temperature increase already after 360 s, independently of the actual power distribution. Such a result could make the assumption of uniform power reasonable for the $\vartheta_{\infty }$ estimates. Greater discrepancies could arise in the temperature increase after a shorter time interval (i.e., for ϑ). However, in all the investigated cases, the length of the selected sequences was in the order of some minutes and the ϑ values remained far below the stated limit, making the conclusions reasonably acceptable.

The separate risk assessments for RF and GC fields disregarded their combined effects. A recent study [22], specifically devoted to joint RF and GC heating in hip implants, showed that the RF and GC thermal effects are maximized for different positions of the implant in the scanner. Thus, the worst cases never coincide, and the maximum cumulative temperature increase tends to be equal to the highest temperature increase between those produced separately by the two sources. Moreover, RF fields usually create hot-spots of SAR near sharpened parts of the implants (e.g., screws, tip of the stem), whereas the coupling with the GC fields is highly relevant when the implant exhibits a large cross section [23].

The use of an ASTM-like phantom opens the very sensitive problem about the relation between in vitro and in vivo results [35]. Previous work showed that the RF heating of the ASTM phantom is usually conservative with respect to the anatomical model, but sometimes, it greatly underestimates the thermal effects [36]. For the GC frequencies, Arduino et al. [34], analysing two anatomical models with hip, knee, and shoulder prostheses, found that a phantom with ASTM gel thermal properties always underestimated the temperature increase with respect to the in vivo results. For this reason, the phantom was filled with a medium having the thermal properties of the bone, including perfusion (which plays an important role in steady-state thermal results) in order to try to better reproduce the local conditions of the implant in the human body. However, the large variability in the thermal effects (up to a factor 2) [37] between different phenotypes makes the reference to in vivo results not unique. In this sense, the phantom seems to be the most general reference, despite the fact that the associated results should be understood as indications. However, in many tier 1 scenarios, the large margin with respect to the stated safety limits may make the evaluation reliable also in vivo.

The validity of the results presented is limited to the investigated scenarios, including the shape and size of the studied implants, the design of the simulated RF coils, and the parameters of the selected MRI pulse sequences. The risk classification could be extended to further scenarios with new implants and pulse sequences, thanks to the availability of a software tool used for processing the simulation results [21].

## 5. Conclusions

This work aimed to provide a better understanding of the heating risk in patients with hip, knee, or shoulder arthroplasty undergoing MRI exams in realistic clinical conditions. A numerical procedure was designed and applied to investigate a large number of clinical scenarios. The results for the RF heating showed that the limits for the whole-body SAR were never exceeded. The limits for the local SAR were exceeded in about 7.5% of the investigated cases only, and the most severe situation (tier 3) was observed in about 3% of the scenarios. The thermal effects produced by GC were found to be limited (tier 3 never occurred).

## Figures and Tables

**Figure 1 diagnostics-12-01873-f001:**
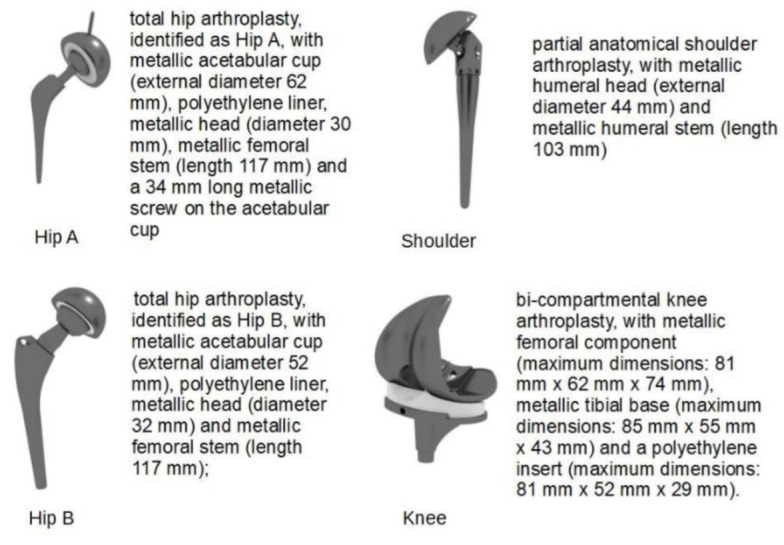
Prosthesis models used in the in silico heating evaluation.

**Figure 2 diagnostics-12-01873-f002:**
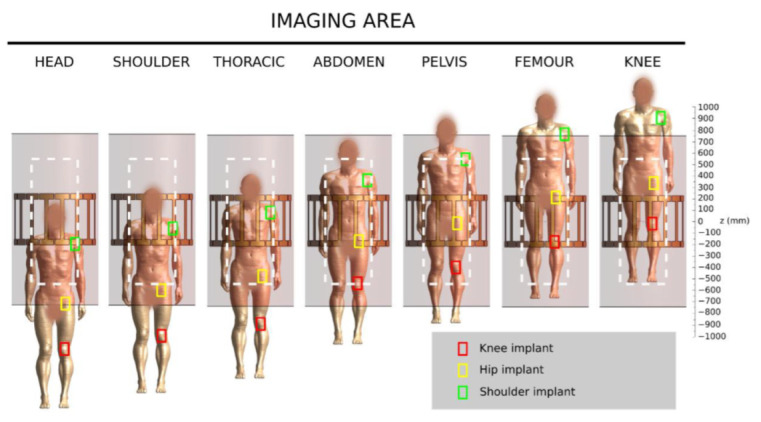
Positions of the human body with respect to the RF body coil and the GC (dark grey rectangle) for the considered imaging zones. The simulated implant positions are represented by the coloured rectangles and the phantom is represented by the dashed white rectangle.

**Figure 3 diagnostics-12-01873-f003:**
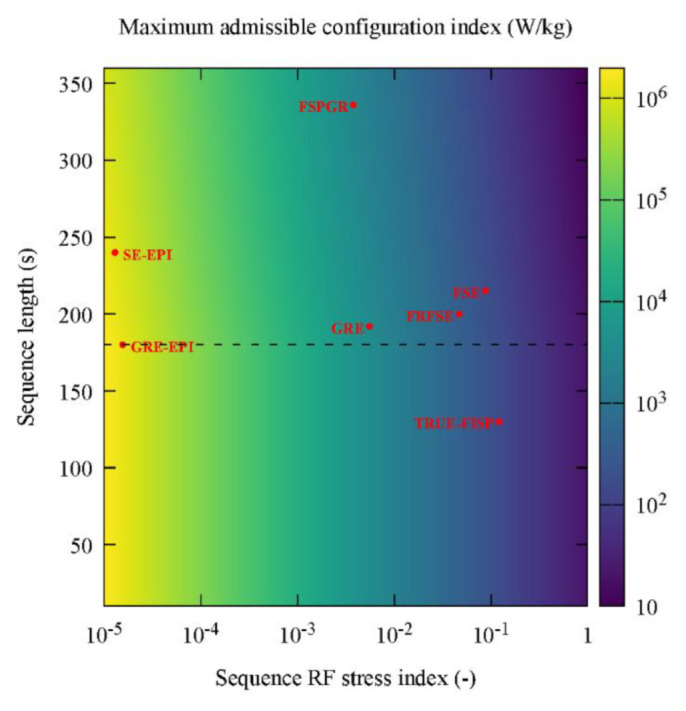
Chromatic map of maximum admissible configuration RF SAR index $\xi_{TR,Max}$ as a function of the RF stress index and of the sequence duration for hip and shoulder implants. The corresponding points in the *ψ**_A_*-*T_seq_* plane of the used sequences are reported. The dashed line represents a sequence duration equal to T_IEC_/2. Below this value, $\xi_{10g,S}^{*}$ becomes independent of the sequence duration.

**Figure 4 diagnostics-12-01873-f004:**
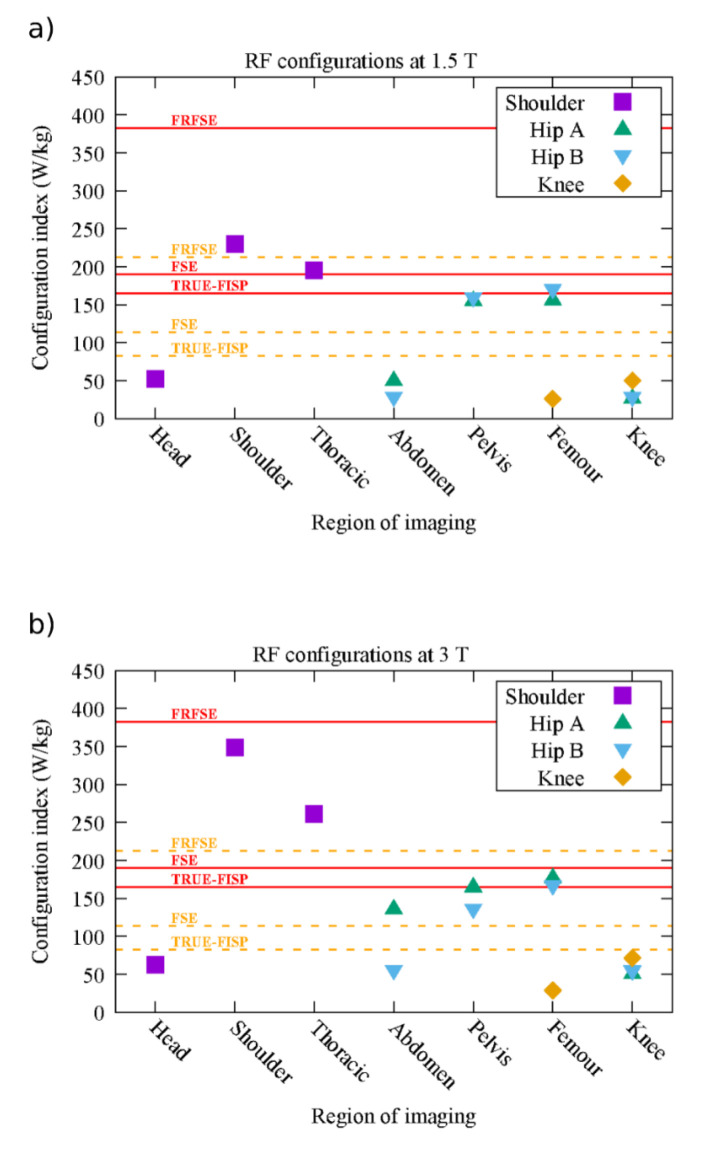
Configuration index $\xi_{10g,C}$ for the most stressed situations at 1.5 T (**a**) and 3 T (**b**). The maximum values of this index ($\xi_{10g,Max}$) admissible for condition (1) and (2) associated with the TrueFISP, T1 FSE, and T2 FRFSE sequences are reported as solid lines (condition 1) and dotted lines (condition 2). For compatibility reasons, the values of $\xi_{10g,C}$ for knee are fictitiously halved.

**Figure 5 diagnostics-12-01873-f005:**
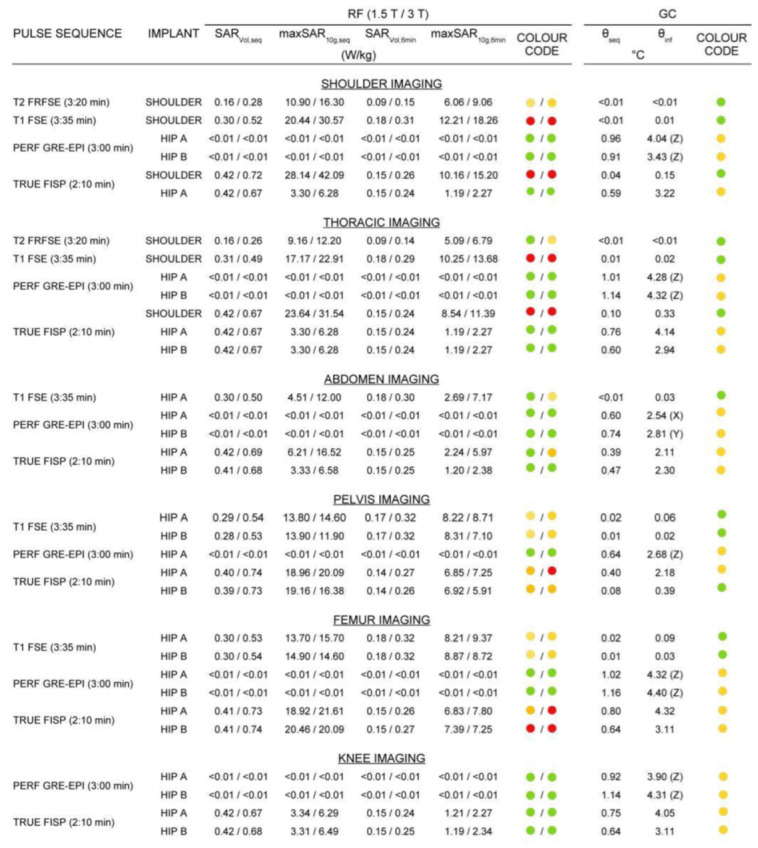
Radiofrequency (RF) and gradient coil (GC)-heating stress evaluation for the scenarios of risk tier 3 or tier 2. Tiers 3, 2 and 1 are associated with red, yellow and green colours, respectively.

**Figure 6 diagnostics-12-01873-f006:**
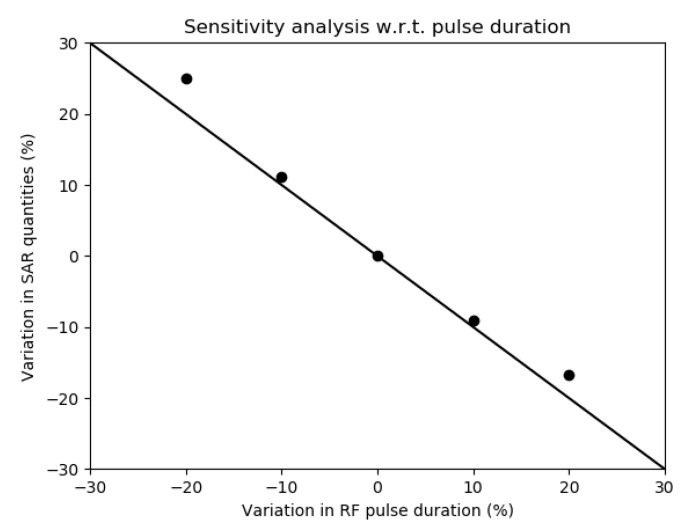
Relative variation in the estimated SAR quantities against relative variation in the RF pulse duration when the time–bandwidth product is kept constant. The result of this sensitivity analysis is independent of the reference sequence.

**Table 1 diagnostics-12-01873-t001:** Parameters relevant to the analysed MRI pulse sequences.

	T2 FRFSE	T1 FSE	T2* GRE	3D FSPGR	DWI SE-EPI	PERF GRE-EPI	TrueFISP
Flip angle (°)	90	90	20	12	90	90	45
B_1_^+^ rms (µT)	1.27	1.74	0.44	0.36	0.02	0.02	2.05
TE (ms)	68	Min (13.2)	15	6	Min (115.2)	Min (50.8)	3.2
TR (ms)	3160	840	500	14.6	5625	1500	6.4
FOV (mm^2^)	300 × 300	300 × 300	300 × 300	300 × 300	300 × 300	300 × 300	120 × 180
Slice thickness (mm)	5	5	5	1	5	5	4
Matrix	256 × 256	256 × 256	256 × 256	256 × 128	128 × 128	128 × 128	256 × 256
Readout BW (kHz)	20.83	31.25	15.63	27.78	250	250	126.3
Specific parameters	Echo train = 16	Echo train = 4	-	Slab thickness = 180 mm	3 directions, b = 500 s/mm^2^	-	-
Duration (min:s)	03:20	03:35	03:12	05:36	04:00	03:00	02:10
*ψ_S_*	4.7 × 10^−2^	8.8 × 10^−2^	5.5 × 10^−3^	3.8 × 10^−3^	1.3 × 10^−5^	1.6 × 10^−5^	1.2 × 10^−1^

**Table 2 diagnostics-12-01873-t002:** Types of MRI exams performed within three years following the surgery for the patients with hip, knee, or shoulder prostheses implanted in 2013.

	Prosthesis							
	Hip		Knee		Shoulder		All Three Prostheses	
MRI Exam	Number of Exams	% of MRI Exams	Number of Exams	% of MRI Exams	Number of Exams	% of MRI Exams	Number of Exams	% of MRI Exams
head	390	19.8%	361	19.4%	26	16.5%	777	19.5%
chest	34	1.7%	25	1.3%	3	1.9%	62	1.6%
spine	749	38.0%	716	38.5%	51	32.3%	1516	38.0%
musculoskeletal	641	32.6%	607	32.7%	68	43.0%	1316	33.0%
abdomen/pelvis	125	6.3%	119	6.4%	9	5.7%	253	6.3%
other	30	1.5%	31	1.7%	1	0.6%	62	1.6%
total	1969	100.0%	1859	100.0%	158	100.0%	3986	100.0%

## Data Availability

Data available in a publicly accessible repository: https://doi.org/10.5281/zenodo.4388310 (accessed on 27 July 2022).

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
