# Peer review of "Classification Scheme of Heating Risk during MRI Scans on Patients with Orthopaedic Prostheses"

_diagnostics, 2022, doi:10.3390/diagnostics12081873_

Round 1

Reviewer 1 Report

Review

Classification scheme of heating risk during MRI scans on patients with orthopaedic prostheses

The paper addresses and interesting research subject. In fact, there is no reliable heating risk assessment when patients with arthroplasty undergo MRI scans. The authors propose a simple procedure to estimate the thermal effects induced in patients with different types of prosthesis. The results are interesting and show that the whole body specific absorption rate is not a self-reliant safety parameter for patients with metallic implants and that less than 10% of the clinical scenarios should require some caution.

The study described is an excellent piece of research work and was carefully performed, adding a new approach to the problem of heating of metallic implants when patients undergo MRI.

The Introduction section is suitable for the purpose of the study. The Materials & Methods section is adequately described, particularly the mathematics of the RF and GC models. The prosthesis models selected (scenario selection) are also adequate. It would be interesting if a titanium prosthesis could have been included in the study to prove that these type of materials produce lower temperature increases. Although reference [24] was cited. The heating risk classification used is suitable. Correct at line 235 –“3. Finally, a safe situation…”.

The number of exams analysed is significant and gives total confidence to the results obtained and respective discussion. As for the Discussion section, it was a correct decision separating the discussion considering the frequency analysis and selected scenarios, RF heating and GC heating. The results are obviously limited to the scenarios chosen and I strongly advise to extend the risk classification to other implants.

 Paper should be accepted for publication.

Author Response

We would like to thank the reviewer for the flattering comments and for the suggestions which has been pursued.

In particular, we added in the paper a quantitative detail about the comparison of the temperature increase between CoCrMo and titanium alloys when radiated by GC fields. 

Reviewer 2 Report

diagnostics-1832548

Title: Classification Scheme of Heating Risk During MRI Scans on Patients with Orthopaedic Prostheses

This manuscript investigated the heating risk from RF and gradient which is quite interesting and would be useful for the community as guidance.

[Comments]

  • How do the gradients contribute to the heating? 
  • In page 2, line 72: how did the authors directly assess the temperature increase?
  • In page 4, line 148, “The thermal problem” Is this also the case in practice?
  • What exactly is the frequency analysis?
  • How did the authors create the simulation model of the implants together with human body parts?
  • Please provide the name of the scanner manufacturer.
  • In page 8, 3.2 RF heating: Please clarify what are the results from the simulations and from the real experiments.
  • In page 13, line 348: This is hardly possible. Please investigate further for the reason of this.
  • Please explain why the different sequences provide different levels of heating.

Author Response

We would like to thank the reviewer for his comments and suggestions.

Below we provide an answer to the raised issues (in red):

How do the gradients contribute to the heating? 

The GC field, like all the time-varying magnetic fields, induces eddy currents inside both the conductive tissues and metallic components of the implants. However, the heating due to the former results to be negligible as a consequence of the low frequency values and tissues conductivity, differently from the RF scenario. Therefore, when GC fields are involved, the metallic implant heats up and then the heat diffuses into the surrounding tissues causing their temperature increase. For the sake of precision, we added a small modification in the paper (line 65). 

In page 2, line 72: how did the authors directly assess the temperature increase?

The reviewer is right. Indeed, we realise that the term "directly" could be misinterpreted. We meant that for GC heating we focused on the final heating effect which is the temperature increase. Of course, the temperature increase has been computed from the deposited power through a numerical solution of the thermal problem. For the sake of clarity, we removed the term "directly" from the paper.

In page 4, line 148, “The thermal problem” Is this also the case in practice?

A uniform power distribution within the implant represents an assumption that allowed us to strongly simplify the numerical procedure without significantly affecting the overall reliability of the results. The relevant limitations are deeply discussed in the "4.4 Limitations" section.

What exactly is the frequency analysis?

In our paper the frequency analysis aimed at defining which implants were mostly scanned and for which purpose. This has been carried out to define the most common scenarios. 

How did the authors create the simulation model of the implants together with human body parts

The CAD models of the shoulder, knee and hip implant B have been provided by the manufacturer who has been cited in the acknowledgement section. Hip implant A has been obtained through 3D scanning of an existing implant. This information is now added to the paper.
In this paper, all the implants have been positioned inside an ASTM-like phantom and no human model have been accounted for.

Please provide the name of the scanner manufacturer.

Being our research fully based on numerical simulations, no specific scanner has been involved. However the geometry of the RF body coil was compliant with the body coil of a Siemens VERIO scanner. The GC coils are compliant with the set of GC coils available in our laboratory for GC measurement. The characteristic of such coils are comparable to those of the GC coils coming along the most common whole body MRI scanners.

In page 8, 3.2 RF heating: Please clarify what are the results from the simulations and from the real experiments.

All the results come from numerical simulations (virtual experiments) as specified in the paper introduction. No laboratory experiment has been presented in our paper.

In page 13, line 348: This is hardly possible. Please investigate further for the reason of this.

In Figure 10 of reference [23] it is shown how the temperature increase of a cylindrical metallic object positioned inside an ASTM phantom is influenced by its length together with the electric properties of the background. Generally, for the examined cases, the temperature increases at 3 T were higher than those at 1.5 T whenever the object length was below its resonance at 3 T. For larger lengths, the heating at 1.5 T generally overcame that at 3 T. We added the proper citation in that point of the paper.

Please explain why the different sequences provide different levels of heating.

Both the RF and GC heating phenomena are regulated by the Faraday's law of induction and, hence, by the time derivative of the magnetic field.

As regards RF, the number, intensity and duration of pulses and repetition time is different for each sequence. This reflects on the stress index ψs and, therefore, on the heating level.
As regards GC, the gradient waveform is sequence dependent and, therefore, the subsequent heating depends on the sequence.

We made an addition in the paper at the beginning of section 2.1 to clarify this point.